# Assessment of Within- and Inter-Patient Variability of Uremic Toxin Concentrations in Children with CKD

**DOI:** 10.3390/toxins16080349

**Published:** 2024-08-09

**Authors:** Evelien Snauwaert, Stefanie De Buyser, An Desloovere, Wim Van Biesen, Ann Raes, Griet Glorieux, Laure Collard, Koen Van Hoeck, Maria Van Dyck, Nathalie Godefroid, Johan Vande Walle, Sunny Eloot

**Affiliations:** 1Ghent University Hospital, 9000 Ghent, Belgium; an.desloovere@uzgent.be (A.D.); wim.vanbiesen@ugent.be (W.V.B.); ann.raes@uzgent.be (A.R.); griet.glorieux@ugent.be (G.G.); johan.vandewalle@ugent.be (J.V.W.); sunny.eloot@ugent.be (S.E.); 2Biostatistics Unit, Faculty of Medicine and Health Sciences, Ghent University, 9000 Ghent, Belgium; stefanie.debuyser@ugent.be; 3CHC Liège, 4000 Liège, Belgium; laure.collard@chc.be; 4Antwerp University Hospital, 2650 Antwerp, Belgium; koen.vanhoeck@uza.be; 5University Hospital Leuven, 3000 Leuven, Belgium; maria.vandyck@uzleuven.be; 6University Hospital Saint Luc, 1200 Brussels, Belgium; nathalie.godefroid@ucl.be

**Keywords:** child, uremic toxins, chronic kidney disease, diet, fiber, protein intake

## Abstract

To promote improved trial design in upcoming randomized clinical trials in childhood chronic kidney disease (CKD), insight in the within- and inter-patient variability of uremic toxins with its nutritional, treatment- and patient-related confounding factors is of utmost importance. In this study, the within- and inter-patient variability of a selection of uremic toxins in a longitudinal cohort of children diagnosed with CKD was assessed, using the intraclass correlation coefficient (ICC) and the within-patient coefficient of variation (CV). Subsequently, the contribution of anthropometry, estimated glomerular filtration rate (eGFR), dietary fiber and protein, and use of (prophylactic) antibiotics to uremic toxin variability was evaluated. Based on 403 observations from 62 children (median seven visits per patient; 9.4 ± 5.3 years; 68% males; eGFR 38.5 [23.1; 64.0] mL/min/1.73 m^2^) collected over a maximum of 2 years, we found that the within-patient variability is high for especially protein-bound uremic toxins (PBUTs) (ICC < 0.7; within-patient CV 37–67%). Moreover, eGFR was identified as a predominant contributor to the within- and inter-patient variability for the majority of solutes, while the impact of the child’s anthropometry, fiber and protein intake, and antibiotics on the variability of uremic toxin concentrations was limited. Based on these findings, we would recommend future intervention studies that attempt to decrease uremic toxin levels to select a (non-dialysis) CKD study population with a narrow eGFR range. As the expected effect of the selected intervention should exceed the inter-patient variability of the selected uremic toxins, a narrow eGFR range might aid in improving the trial design.

## 1. Introduction

In chronic kidney disease (CKD), toxic organic metabolites, so called “uremic toxins”, accumulate in parallel with the deterioration of kidney function. Uremic toxins are classically divided based on their physicochemical characteristics explaining their elimination during dialysis into three categories: small, water-soluble compounds; middle molecules; and protein-bound uremic toxins (PBUTs) [1]. Uremic toxins contribute to the systemic morbidity and mortality present in CKD next to several other factors that contribute to the systemic nature of CKD, i.e., the disturbances in the endocrine and homeostasis functions of the kidney, and symptoms related to the native kidney disease and its treatment [2]. 

Previously published meta-analysis data assessing the possible impact of dietary and/or pharmacological intervention on uremic toxin concentrations, and with it, their biological toxicity, have often shown conflicting results without providing strong conclusions [3,4,5,6,7,8,9,10]. The conflicting results have been explained by several factors such as the quality of trial design, the low number of trials, and the variability in study populations due to confounding factors such as nutritional aspects, treatment- and patient-related factors. 

To halt the unacceptably high morbidity and mortality in patients with CKD that is partially caused by uremic toxin accumulation, new treatments that can mitigate the toxicity of uremic toxin accumulation are urgently needed. To promote improved trial design in upcoming RCT’s assessing new treatments, insight in the within- and inter-patient variability of uremic toxins contributing to these nutritional, treatment- and patient-related confounding factors is of utmost importance. A previous study in 18 adult patients on maintenance hemodialysis found substantial within- and inter-patient variability of pre-dialysis concentrations of several uremic toxins over a short follow-up period of 16 weeks [11]. Nevertheless, these results cannot be generalized to the non-dialysis CKD population in which variability of kidney function is considered an additional important contributor to the accumulation pattern of uremic toxins. Clinical studies investigating the within-patient variability of uremic toxins in the non-dialysis CKD population are inexistent. Moreover, no data are available on uremic toxins’ within- and interpatient variability in specific populations such as in children, who are known to differ in body composition (i.e., larger body water composition), diet pattern (i.e., higher caloric/protein requirements per kg body weight), microbiome composition (only stable from the age of 3–5 years), and concentrations of circulating proteins in comparison to adults [12]. 

Therefore, we aim to assess within- and inter-patient variability of a selection of uremic toxins in a longitudinal prospective cohort of children with CKD stage 1–5 (non-dialysis). Second, we aim to evaluate the impact of patient- and treatment-related characteristics such as patient’s anthropometry, eGFR, dietary fiber and protein, and use of antibiotic prophylaxis on the within- and inter-patient variability of uremic toxin concentration. 

## 2. Results

The dataset contained 403 observations coming from 62 subjects, with a median of 7 visits per patient over the maximum follow-up time of 2 years. Complete data at entry of the study on dietary protein and fiber, antibiotics use, body surface area (BSA), and eGFR are summarized in Table 1. 

At entry in the study, participants had a mean age of 9.4 ± 5.3 years, were predominantly males (68%) and 61% of children had CKD stage 1–3. Moderate to severe CKD stages 4–5 was present in 39% of children, and 15% underwent a kidney transplantation prior to entry into the study. Congenital anomalies of the kidney and urinary tract were the predominant underlying renal diagnosis in this pediatric cohort. In total, 28% were on antibiotic prophylaxis at entry of the study, i.e., 14 out of 17 were on trimethoprim or (nitro) furantoin antibiotic prophylaxis. At entry, the median dietary protein and fiber intake was, respectively, 52.8 and 11.7 g/m^2^/day. The median protein/fiber index in our cohort was 4.4.

The within-patient variability in uremic toxins is displayed in Table 2 by means of the intraclass correlation coefficient (ICC) and within-patient coefficient of variation (CV). The lowest ICCs (<0.7) were found for urea, asymmetric dimethyl-arginine (ADMA), hippuric acid (HA) and indoxyl sulfate (IxS), corresponding to a higher within-patient variability or a lower inter-patient variability (Table 2, Figure 1). For HA and IxS, the low ICCs are associated with high within-patient CV (79% and 46%, respectively). In contrast, the low ICCs from urea and ADMA are especially attributed to a low inter-patient variability. The pre-set ICC threshold of >0.7 was obtained in all studied middle molecules (0.898 and 0.959 for, respectively, β_2_-microglobulin (β_2_M) and complement factor D (CfD)), reflecting a low within-patient variability (as confirmed by the concomitant low within-patient CV (13–27%)). Similar high ICCs were also obtained for small water-soluble solutes, as for symmetric dimethyl-arginine (SDMA) (0.881), and uric acid (UA) (0.909), with a corresponding low within-patient CV of 13–24%.

The percentage of within- and inter-patient variance in uremic toxins that could be explained by dietary protein, fiber, antibiotic use, eGFR, BSA, or the combination of all five, is summarized in Table 3. The combination of all five variables could only explain the within-patient variance of IxS (6%), and β_2_M (11%). Dietary protein, fiber, and antibiotic use did not help explain the within-patient variance of the selected uremic toxins. eGFR explained the within-patient variance of indole acetic acid (IAA) (5.1%), IxS (6.0%), β_2_M (12%) and CfD (6.3%) (Table 3).

With the exception of ADMA and CMPF, the inter-patient variance of the selected uremic toxins was for 15 to 82% explained by the combined model, of which eGFR was—as expected—the predominant contributor to the present inter-patient variance. The proportion of the explained inter-patient variance was lower for PBUTs (i.e., ranging from 21% for pCS to 66% for IxS) in comparison to small water-soluble molecules and the middle molecules (i.e., ranging from 67% for SDMA up to 79% for urea). 

## 3. Discussion

This study assessed and characterized the within- and inter-patient variability of a selection of uremic toxins in a longitudinal pediatric cohort with (non-dialysis) CKD stage 1–5. In the present study, we found (i) that the within-patient variability is high for especially the PBUTs HA and IxS; (ii) that eGFR is the predominant contributor to the within-patient variability of IAA, IxS, β_2_M and CfD and of the inter-patient variability for the majority of selected solutes; and (iii) that the impact of the child’s anthropometry, dietary and protein intake, and the use of antibiotics on the variability of uremic toxin concentrations is limited. This is one of the largest longitudinal pediatric cohorts assessing within- and inter-patient variability of a selection of uremic toxins. 

First, a high within-patient variability of PBUTs, most prominently present for HA and IxS, was observed. Our study also highlights the minimal within-patient variability present in the assessed middle molecules (β_2_M, CfD) and small water-soluble uremic solutes (SMDA and UA). Similar low ICCs were reported by Eloot et al. [11] in an adult hemodialysis population for HA, IxS and ADMA. In contrast to the study of Eloot et al. [11], we could not find low ICCs for SDMA and UA, which might be related to the lower inter-patient variability present in the cohort of Eloot et al. that included only adults on hemodialysis during a short follow-up time (i.e., 4 months) while our pediatric cohort included a more heterogenous group of children with different ages, variable degree of kidney impairment, and a longer follow-up interval of up to 2 years. 

Second, we found that eGFR is the predominant contributor to the within-patient variability of IAA, IxS, β_2_M and CfD and the inter-patient variability of the majority of selected solutes. The importance of the kidney in explaining the accumulation pattern of uremic toxins has also been shown by several other studies. For instance, Gryp et al. [13] found that the limited excretions of PBUT by the kidneys predominantly explained the accumulation pattern of PBUTs. Similarly, previous reported studies in adult and pediatric dialysis cohorts demonstrated that levels of both middle molecules and PBUTs were correlated to residual kidney function. 

Third, we found that children’s dietary fiber and protein intake, and use of antibiotic prophylaxis did not help explain the within- and inter-patient variability of uremic toxin levels in a pediatric CKD cohort not on dialysis and with a follow-up of at max 2 years. The role of diet and antibiotics is especially relevant in the context of PBUTs, as these solutes are predominantly derived from amino acid metabolism by the intestinal bacteria. Changes in nutrient intake are hypothesized by several studies to contribute to the variability of especially PBUT accumulation, as both dietary fiber and protein consumption were previously associated with PBUT plasma levels in children and adults with CKD [14,15]. Moreover, several studies have suggested that the differences in bacterial generation that occur alongside the deterioration of kidney function contribute to the variability present in gut-derived PBUTs plasma levels [15,16,17,18,19]. Nevertheless, the presence of increased bacterial generation could not be confirmed by others, for instance, Gryp et al. [13] detected no differences in bacterial generation in the gut between mild and advanced stages of CKD. Studies assessing the impact of antibiotic therapy on PBUTs plasma levels have also been conflicting. While the study of Eloot et al. [11] did not find a difference in ICC values between patients with or without antibiotic therapy (i.e., temocillin, amoxicillin ± clavulanic acid, vancomycin), Nazzal et al. [20] have shown acute changes in PBUT levels after initiating oral vancomycin. However, the results on antibiotic use obtained in the present study have to be interpreted with caution when comparing with these studies. In the present cohort, patients were sampled away from acute illnesses and broad-spectrum antibiotic therapies and the antibiotic use was restricted to prophylactic antibiotic use only. 

We found that the child’s anthropometrics only had a minor contribution to the inter-patient variability of UA. Although several maturational and developmental processes occur through childhood that might hypothetically impact the generation (i.e., ongoing intestinal microbiota development until the first 2–3 years of life; high protein requirements per kg body weight), multicompartmental distribution and intercompartmental shifts (i.e., lower circulating plasma proteins, larger body water volumes proportionally), and the excretion (i.e., increase in organic solute transport in first 2 years) of uremic toxins, we were not able to explain variability by the child’s anthropometrics. 

At last, important to note is that, while the % explained inter-patient variability was high (77–92%) for the small water-soluble compounds (urea, SDMA) and middle molecules, the total % explained inter-patient variability for the PBUTs for the selected contributors was only 15–67%. This, in combination with the overall low % explanatory within-patient variability found in this study, suggests the presence of other contributors to the within- and inter-patient variability of (especially protein-bound) uremic toxins. For instance, the preserved active tubular function is known to contribute to PBUTs accumulation and is only partially reflected by eGFR [21]. Also the variability in PBUTs precursors by gut microbial production is not assessed in this cohort. At last, 15% of children received a kidney transplant prior to entry of the study, of which is known that the accumulation pattern of especially PBUTs is different in non-transplanted versus transplanted patients. For instance, lower IxS levels were found in the transplant cohort in comparison to the non-transplant cohort, and persistent changes in the microbiota after transplantation are also described [22,23,24]. Additional efforts are needed to further explore the impact of factors other than eGFR on the within- and interpatient variability of especially PBUTs.

Whereas this study is the first to assess the within- and interpatient variability of uremic toxins in children with CKD, our study has also limitations that need to be addressed. First, the heterogenicity of the rather small cohort of children might have hampered us in finding the effect of diet and antibiotics on uremic toxin variability. Indeed, the cohort included children with diverse types of kidney disease including post-kidney transplantation, different age categories and a wide range of kidney impairment. Second, we assumed that the dietary habits of patients were constant and incorporated only one single dietary measurement throughout the follow-up in the analysis. Although we acknowledge that recall biases by the use of 24 h recalls in our design cannot be excluded, the incorporation of the standardized 24 h recall alternative has allowed us to balance the inherent disadvantages of 3 days food records in this observational study (i.e., high burden, incomplete recording, necessity of literacy skills). While only a few studies incorporated detailed dietary information in the description of uremic toxin accumulation, we acknowledge that this might have reflected an incomplete view on the dietary aspects in the uremic toxin accumulation pattern. 

## 4. Conclusions

In conclusion, we found that within-patient variability is present for especially PBUTs. Moreover, we demonstrated that eGFR is a predominant factor contributing to within/inter-patient variability, while variability could not be explained by dietary intake, antibiotic use and the child’s anthropometrics. Moreover, we demonstrated that the total % explanatory within- and inter-patient variability of the PBUTs is low, which suggests the presence of other contributors to its variability. Based on these findings, we would recommend future intervention studies that attempt to decrease uremic toxin levels to select a (non-dialysis) CKD study population with a narrow eGFR range. As the expected effect of the selected intervention should exceed the inter-patient variability of the selected uremic toxins, a narrow eGFR range might aid in improving the trial design. 

## 5. Materials and Methods

### 5.1. Patients

Children (0–18 years) diagnosed with CKD stage 1-5D were recruited between August 2014 and December 2017 from Ghent University Hospital, University Hospital Antwerp, University Hospital Leuven, CHC Liège and University Hospital Saint-Luc Brussels for prospective longitudinal follow-up every 3 months (up to 24 months). CKD was defined according to the KDIGO guidelines and classified into different stages (1 to 5D) according to eGFR, determined by the updated bedside Schwartz eGFR equation (Figure 2) [25]. Exclusion criteria were active systemic inflammatory diseases (e.g., systemic lupus erythematosus, bone marrow transplantation) or active malignancy (e.g., posttransplant lymphoproliferative disease and malignancy under chemotherapy). Visits were planned away from active bacterial or viral infectious diseases (e.g., urinary tract infections, respiratory infections) with implications for the child’s wellbeing. Visits at which children were receiving any type of dialysis were also excluded from the analysis. The study protocol was approved by the Ethics Committee and written informed consent was obtained from all individual participants included in the study and/or from their parents (B670201524922; B670201422206; ID number Clin.gov: NCT02624466).

### 5.2. Data Collection and Biochemical Measurements

From each participant, one EDTA plasma and one serum were drawn during a routine ambulatory visit (frequency once every 3 months). For a detailed methodology of the biochemical analysis of IxS, IAA, pCS, pCG, HA, CMPF, SDMA, ADMA, CfD, and β_2_M, we refer to the publication of Snauwaert et al. [26]. 

### 5.3. Dietary Assessment

The methodology of dietary assessment is explained in detail by El Amouri et al. [27]. In short, structured 3-day diary templates were completed every 3 months, 3 days prior to the visit and reviewed by a trained dietician in face-to-face interviews. When 3-day diaries were forgotten, the dietary assessment was substituted by a 24 h recall (aiming for a standardized time schedule between dietary data and plasma levels). In order to minimize the impact of the known shortcomings of 24 h recalls (i.e., interview bias, inaccurate estimation of portion size), the 24 h recalls were performed according to a detailed protocol, i.e., by a selected number of skilled and trained dieticians with standardized food models and a food photo album (Portiegroottes boek, Valetudo Consulting, 3rd edition, March 2014) were utilized, along with a manual for the conversion of household measures to weight equivalents [28,29]. Additional information on the analysis of protein and fiber intake is present in the publication of El Amouri et al. [14,27].

### 5.4. Statistical Analyses

Descriptive data are expressed as geometric mean ± standard deviation (SD) or median [25th; 75th percentile], as appropriate. The normality of distribution was checked with both histograms to assess the distribution and the Shapiro–Wilk test. Absolute and relative frequencies are reported for categorical variables. Linear mixed models for log-transformed toxin concentrations were fitted with a random intercept for the patient. In case zero values were registered for a toxin concentration, the measurement error was added to all observations for that respective toxin concentration to allow for log transformation; i.e., 0.00324 units for HA, 0.00086 for pCG, and 0.00001 units for CMPF. 

First, an empty model was fitted to provide important preliminary information about the within-subject (residual) and inter-patient (random intercept) variance in the outcome, which could be explained by including level-1 and level-2 explanatory variables in a multilevel model. The reproducibility of repeatedly measured toxin concentrations over time was assessed. We assumed that the included patients were stable over time. The intraclass correlation coefficient (ICC) [−] was calculated by dividing the inter-patient variance by the total variance, as a measure of within-patient variability. A threshold of ICC > 0.7 was assumed [30], corresponding to a within-patient variance maximum of 65% of the inter-patient variance. The standard error of measurement (SEM) was estimated by taking the square root of the within-patient variance. This standard deviation on the log scale is a dimensionless ratio. If we subtract one from this ratio, we obtain the ratio of the standard deviation to the mean, which is the within-patient coefficient of variation (CV) [31].

Second, six adjusted models were specified: (i) adjusted for dietary protein only (first available measurement, time-fixed), (ii) adjusted for dietary fiber only (first available measurement, time-fixed), (iii) adjusted for antibiotics use only (time-varying), (iv) adjusted for body surface area (BSA) only (time-varying), (v) adjusted for estimated glomerular filtration rate (eGFR) only (time-varying) and (vi) adjusted for dietary protein, dietary fiber, eGFR, BSA and antibiotics use. Each adjusted model was compared to the empty model using a likelihood ratio test. In case the adjusted model fitted the data better than the empty model (*p* < 0.05), the proportional reductions in estimated variances were computed as analogs to R^2^ for multilevel models.

Using Raudenbush and Bryk’s method in multilevel models, the proportion of explained within-patient outcome variation is estimated as the proportional reduction in residual variance comparing the specified model with the empty model [32]. A correctly specified model should reduce the unexplained level-1 variance significantly. Likewise, the proportion of explained within-patient outcome variation is estimated as the proportional reduction in random intercept variance comparing the specified model with the empty model. Using this proportional reduction in estimated variances may encounter problems, such as negative values of ‘R^2^’. It is possible for the variance to decrease when explanatory variables are added. A small decrease may be a result of chance effects. A decrease of >5% would suggest a possible misspecification of the model. The linear mixed models were fitted using the ‘lme4’ package in R. All hypothesis tests were exploratory in nature and performed at the two-sided 5% significance level. No correction for the type I error rate was made. 

## Figures and Tables

**Figure 1 toxins-16-00349-f001:**
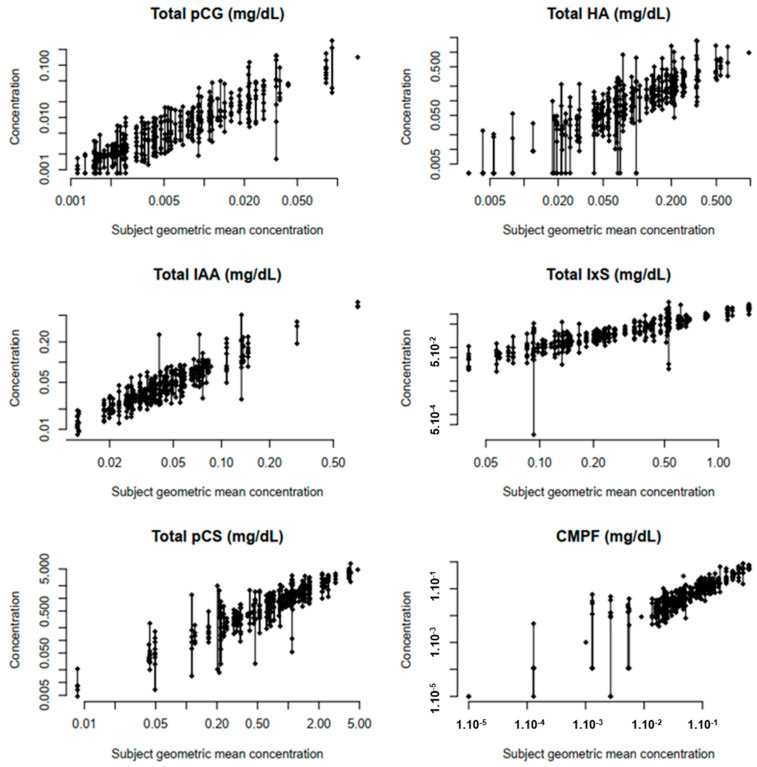
Plots of the observed protein-bound toxin concentrations in the function of ascending within-subject geometric mean toxin concentrations. Abbreviations: HA: hippuric acid; IAA: indole acetic acid; IxS: indoxyl sulfate; PCS: p-cresyl sulfate; CMPF: 3-carboxy-4-methyl-5-propyl-2furanpropionic acid.

**Figure 2 toxins-16-00349-f002:**
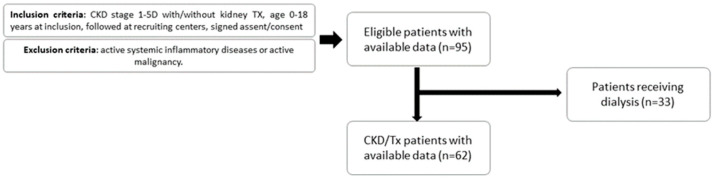
Study flow chart. CKD: chronic kidney disease, TX: kidney transplant.

**Table 1 toxins-16-00349-t001:** Demographic data at entry of the study.

Sample size—*n*	62
Total number of visits—*n*	403
Number of visits per patient—*n*	7 [5; 9]
Gender (M/F)—*n* (%)	42 (68%)/20 (32%)
Age (years)	9.4 ± 5.3
eGFR (mL/min/1.73 m^2^)	38.5 [23.1; 64.0]
CKD classes—*n* (%)	
CKD stage 1	6 (10%)
CKD stage 2	12 (19%)
CKD stage 3	20 (32%)
CKD stage 4	16 (26%)
CKD stage 5	8 (13%)
Renal Diagnosis—*n* (%)	
Cystic disease	4 (7%)
CAKUT	31 (50%)
Glomerulonephritis	10 (16%)
Proximal Tubular Disease	5 (8%)
Other or unknown	12 (19%)
Transplantation—*n* (%)	9 (15%)
BSA (m^2^)	1.0 ± 0.4
Prophylactic antibiotics—*n* (%)	
No	45 (73%)
Trimethoprim	8 (13%)
(Nitro) furantoin	6 (10%)
Azithromycin	1 (2%)
Amoxicillin	1 (2%)
Other	1 (2%)
Nutrient intake	
Protein intake (g/m^2^/day)	52.8 [36.6; 67.3]
Fiber intake (g/m^2^/day)	11.7 [8.2; 15.8]
Protein/fiber index °	4.4 [3.0; 6.4]

° In 3 patients protein/fiber index could not be calculated as fiber intake was 0 g/day.

**Table 2 toxins-16-00349-t002:** Uremic toxin concentrations at baseline, within-patient coefficient of variation (CV), and intraclass correlation coefficient (ICC) in pediatric CKD, compared to the ICC as previously measured in adult hemodialysis patients [11].

	Pediatric CKD Patients			Adult HD Patients [11]
	Concentration at Baseline (mg/dL)	Within-Patient CV (%)	ICC	ICC
Small water-soluble solutes				
Urea	73.4 [40.5; 101]	26	**0.66**	0.74
SDMA	0.028 [0.021; 0.046]	24	0.88	**0.69**
ADMA	0.015 [0.013; 0.018]	16	**0.46**	**0.69**
UA	6.94 [5.64; 8.19]	13	0.91	**0.64**
Protein-bound toxins				
PCG	0.007 [0.002; 0.017]	61	0.75	0.86
HA	0.110 [0.052; 0.166]	79	**0.63**	**0.50**
IAA	0.038 [0.027; 0.066]	37	0.79	0.81
IxS	0.235 [0.105; 0.479]	46	**0.64**	**0.63**
PCS	0.826 [0.355; 1.539]	57	0.76	0.79
CMPF	0.043 [0.014; 0.092]	67	0.76	0.94
Middle molecules				
β_2_M	0.562 [0.342; 0.955]	27	0.90	0.76
CfD	0.657 [0.349; 0.914]	15	0.60	n.a.

Concentrations are displayed as median and 25th and 75th percentile. In **bold**, ICC < 0.7. CV: coefficient of variation; ICC: intraclass correlation coefficient; SDMA: symmetric dimethyl arginine; ADMA: asymmetric dimethyl arginine; UA: uric acid; PCG: p-cresyl glucuronide; HA: hippuric acid; IAA: indole acetic acid; IxS: indoxyl sulfate; PCS: p-cresyl sulfate; CMPF: 3-carboxy-4-methyl-5-propyl-2-furanpropionic acid; β_2_M: beta-2-microglobulin; CfD: complement factor D; n.a.: not available.

**Table 3 toxins-16-00349-t003:** Percentage of explained within- and inter-patient variance by 5 explanatory variables (dietary protein, dietary fiber, antibiotic use, BSA, eGFR) and a combination of all five variables in comparison to an empty model.

% of Explained Within-Patient Variance/% of Explained Inter-Patient Variance						
	Dietary Protein	Dietary Fiber	Antibiotic Use	BSA	eGFR	All 5
Small water-soluble solutes						
Urea	-	-	-	-	4.0/79	3.6/81
SDMA	-	-	-	-	−1.7/67	−0.1/78
ADMA	-	-	-	-	-	-
UA	-	-	-	1.1/18	0.4/26	0.5/42
Protein-bound toxins						
pCG	-	-	-	-	1.1/26	2.3/21
HA	-	-	-	-	−1.8/54	−1.6/51
IAA	-	-	-	-	5.1/44	4.7/47
IxS	-	-	-	-	6.0/66	6.0/67
PCS	-	-	-	-	2.1/21	3.2/15
CMPF	-	-	-	-	-	-
Middle molecules						
β_2_M	-	-	-	-	12/69	11/77
CfD	-	-	-	-	6.3/74	4.7/81

Data are expressed in %. In case the adjusted model was not better than the empty model (i.e., likelihood ratio test not significant), data are displayed as ‘-’. Percentages in grey are small decreases as a result of chance effects or increases of <5%. SDMA: symmetric dimethyl arginine; ADMA: asymmetric dimethyl arginine; UA: uric acid; PCG: p-cresyl glucuronide; HA: hippuric acid; IAA: indole acetic acid; IxS: indoxyl sulfate; PCS: p-cresyl sulfate; CMPF: 3-carboxy-4-methyl-5-propyl-2furanpropionic acid; β_2_M: beta-2-microglobulin; CfD: complement factor D.

## Data Availability

The data presented in this study are available on request from the corresponding author due to privacy.

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
