# Peer review of "Assessment of Within- and Inter-Patient Variability of Uremic Toxin Concentrations in Children with CKD"

_toxins, 2024, doi:10.3390/toxins16080349_

Round 1

Reviewer 1 Report

Comments and Suggestions for Authors

Thank you for the opportunity  to review this manuscript.

Since  standardization of  measures of uremic  concentrations its an important  issue  in pediatric population, the present  manuscript  represent an important  contribution  to  nephrology. Some issues,  however, should  be addressed  to improve  the technical  quality  of the research. 

In the introduction section,  briefly mention  the  expected contribution to everyday clinical practice (i.e.  knowing the factors affecting within-patient variability of uremic toxins may improve overall clinical assessment) in order  to highlight the impact  of  your  research impact.

As  15% of  your population  had  already received a transplant,  in the  introduction section,  briefly mention the  expected similarities  between  a  child with  CKD  before and  after  receiving  a  transplant, in order  to  clarify  issues  regarding   your Population heterogeneity.

In  the result  section, add a flowchart  explaining  initial  population selection,  inclusion and  exclusion  criteria  and final  sample  size.

Also,  explain how  you assessed  the normality  of  your  data, as  it is  of  utmost importance  in the selection of  further statistical  test.

In the discussion section, briefly  mention the  expected  development-associated  changes in  uremic  toxins  concentration and  distribution, as your population  age  range  is wide.

Comments on the Quality of English Language

Moderate  English editing needed

Author Response

3. Point-by-point response to Comments and Suggestions for Authors

Comments 1: In the introduction section, briefly mention the expected contribution to everyday clinical practice (i.e. knowing the factors affecting within-patient variability of uremic toxins may improve overall clinical assessment) in order to highlight the impact of your research impact.

Response 1: Thank you for pointing this out. We thank the reviewer for highlighting the insufficiently described clinical impact. Therefore, we have adjusted the Introduction paragraph below:

Page 2, line 45-48:To halt the unacceptable high morbidity and mortality in patients with CKD that is partially caused by uremic toxin accumulation, new treatments that can mitigate the toxicity of uremic toxin accumulation is urgently needed. To promote improved trial design in upcoming RCT’s assessing new treatments, insight in the within- and inter-patient variability of uremic toxins contributing to these nutritional, treatment- and patient-related confounding factors is of utmost importance.”

Comments 2: As 15% of your population had already received a transplant, in the  introduction section, briefly mention the expected similarities between a child with CKD  before and after receiving a transplant, in order to clarify issues regarding your Population heterogeneity.

Response 2: We agree with the reviewer that the proportion of children that received a kidney transplant is not to neglect and might indeed have contributed to the population heterogenicity in this paper. If the reviewer can agree with us, we would like to propose to address this concern to the limitation section of the Discussion.

Page 6, line 195-205: This, in combination with the overall low % explanatory with-in-patient variability found in this study, suggests the presence of other contributors to the within- and inter-patient variability of (especially protein-bound) uremic toxins. For instance, the preserved active tubular function is known to contribute to PBUTs accumulation and is only partially reflected by eGFR [29, 30]. Also the variability in PBUTs precursors by gut microbial production is not assessed in this cohort. At last, 15% of children received a kidney transplant prior to entry of the study, of which is known that the accumulation pattern of especially PBUTs is different in non-transplanted versus transplanted patients. For instance, lower IxS levels were found in the transplant cohort in comparison to the non-transplant cohort, and persistent changes of the microbiota after transplantation are also described[31-33].

Page 7, line 2011:  Indeed, the cohort included children with diverse types of kidney disease including post-kidney transplantation, different age categories and a wide range of kidney impairment.

Comments 3: In the result section, add a flowchart explaining initial population selection, inclusion and exclusion criteria and final sample size.

Response 3: We agree that a study flow chart is missing in our manuscript, and adding this information will enhance the clarity and transparency of our work. Therefore, the study flow chart was added in the manuscript section, as requested. We would like to refer to Figure 3 for visualization of the study flow chart.

Comments 4: Also, explain how you assessed the normality of your data, as  it is of utmost importance in the selection of further statistical test.

Response 4: We apologize for not clarify the method of normality in our method section. We thank the reviewer for pointing out. We added this information to the Method paragraph.

Page 8, line 294-297: Descriptive data are expressed as geometric mean ± standard deviation (SD) or median [25th; 75th percentile], as appropriate. Normality of distribution was checked with both histograms to assess distribution and the Shapiro–Wilk test. Absolute and relative frequencies are reported for categorical variables.

Comments 5: In the discussion section, briefly  mention the  expected  development-associated  changes in  uremic  toxins  concentration and  distribution, as your population  age  range  is wide.

Response 5: We agree that the maturational and developmental processes are not explained to the reader. We have, accordingly, modified the Discussion to emphasize this point. We refer to the paragraph below to demonstrate the modifications:

Page 6, line 189-194: “Although several maturational and developmental processes occur through childhood that might hypothetically impact the generation (i.e. ongoing intestinal microbiota development until the first 2–3 years of life; high protein requirements per kg body weight), multicompartmental distribution and intercompartmental shifts (i.e. lower circulating plasma proteins, larger body water volumes proportionally), and the excretion (i.e. increase in organic solute transport in first 2 years) of uremic toxins, we were not able to explain variability by the child’s anthropometrics.”

5. Additional clarifications

Not applicable

Reviewer 2 Report

Comments and Suggestions for Authors

The present study aims to describe the within- and inter-patient variability of a selection of uremic toxins in a longitudinal prospective cohort of children with (non-dialysis) CKD and to evaluate the impact of patient- and treatment-related characteristics such as patient’s anthropometry, estimated glomerular filtration rate (eGFR), dietary fiber and protein intake, and use of antibiotic prophylaxis on the within- and inter-patient variability of uremic toxin concentrations. The study provides valuable insights into the variability of uremic toxins in children with CKD and highlights the significant role of eGFR. However, it is important to address the potential biases and limitations in dietary assessment and data collection methods.

While the study finds eGFR as the predominant factor contributing to uremic toxin variability, it may overemphasize this finding without adequately exploring the potential impact of other factors.

The authors should address potential selection biases and discuss how representative the cohort is of the general paediatric CKD population.

The confidence on 24-hour dietary recalls as substitutes for 3-day food records introduces potential recall bias, as patients may not accurately remember their intake. The study should discuss any validation or cross-checking measures taken to ensure consistency between the two methods.

The purpose for a 50/50 ratio between 3-day food records and 24-hour recalls might lead to inconsistencies in data quality and comparability.

Collecting dietary intake data every three months might not capture short-term dietary variations that could influence uremic toxin levels. More frequent dietary assessments could provide a complete understanding of the relationship between diet and uremic toxin variability.

Comments on the Quality of English Language

English looks good

Author Response

Comments 1: The present study aims to describe the within- and inter-patient variability of a selection of uremic toxins in a longitudinal prospective cohort of children with (non-dialysis) CKD and to evaluate the impact of patient- and treatment-related characteristics such as patient’s anthropometry, estimated glomerular filtration rate (eGFR), dietary fiber and protein intake, and use of antibiotic prophylaxis on the within- and inter-patient variability of uremic toxin concentrations. The study provides valuable insights into the variability of uremic toxins in children with CKD and highlights the significant role of eGFR. However, it is important to address the potential biases and limitations in dietary assessment and data collection methods.

Response 1: We thank the reviewer for the feedback. We would like to refer to the comments below in which all raised concerns and limitations by the reviewer are addressed.

Comments 2: While the study finds eGFR as the predominant factor contributing to uremic toxin variability, it may overemphasize this finding without adequately exploring the potential impact of other factors.

Response 2: We thank the reviewer for raising this point. Indeed, eGFR has been identified in our study as the predominant factor contributing of both the within as inter-patient variability. As described in the limitation section, the heterogenicity of our rather small pediatric cohort have possibility hampered us to find an effect of diet and antibiotics on variability. We agree that further studies are needed to further explore the potential impact of other factors than eGFR, which is also supported by our data in which only 15-67% of the inter-patient variability could be explained by our composite model including diet, body surface area, antibiotic use and eGFR. We therefore adjusted the following paragraph in discussion accordingly:

Page 7, line 195:At last, important to note is that, while the % explained inter-patient variability was high (77-92%) for the small water soluble compounds (urea, SDMA) and middle molecules, the total % explained inter-patient variability for the PBUTs for the selected contributors was only 15-67%. This, in combination with the overall low % explanatory within-patient variability found in this study, suggests the presence of other contributors to the within- and inter-patient variability of (especially protein-bound) uremic toxins. For instance, the preserved active tubular function is known to contribute to PBUTs accumulation and is only partially reflected by eGFR [29, 30]. Also the variability in PBUTs precursors by gut microbial production is not assessed in this cohort. At last, 15% of children received a kidney transplant prior to entry of the study, of which is known that the accumulation pattern of especially PBUTs is different in non-transplanted versus transplanted patients. For instance, lower IxS levels were found in the transplant cohort in comparison to the non-transplant cohort, and persistent changes of the microbiota after transplantation are also described[31-33]. Additional efforts are needed to further explore the impact of factors other than eGFR on the within- and interpatient variability of especially PBUTs.

Comments 3: The authors should address potential selection biases and discuss how representative the cohort is of the general pediatric CKD population.

Response 3: We thank the reviewer for pointing out this concern. We agree that the selection of patients in our manuscript was not clearly explained in the methodology. To enhance the clarity and transparency of our work with respect to patient selection, we added a study flow chart in the manuscript section (see figure 3). We believe that our cohort contains a representative selection of the general pediatric CKD population. In Belgium, the care for children with CKD is centralized in a limited number of centers with access to a multidisciplinary follow-up program, of which 5 of the total of 6 specialized centers were included in our observational study. The fact that only Belgian centers are included has the advantage that the overall management is quite similar across the centers (i.e. access to growth hormone therapy and support from dietician, access dialysis/transplant, dietary habits, …). Moreover, the inclusion criteria were set broad, without limitation of underlying kidney disease of present kidney function. Only children with active systemic inflammation (such as active systemic lupus erythematosus, bone marrow transplantation) or with active malignancy (under chemotherapy or active post-transplant lymphoproliferative disease) were excluded. As can be appreciated in table 1, the distribution of underlying kidney disease, age, and sex reflects a similar distribution as described in real-life registry data across Europe. We would like to refer to Figure 3 for the added study flow chart.

Comments 4: The confidence on 24-hour dietary recalls as substitutes for 3-day food records introduces potential recall bias, as patients may not accurately remember their intake. The study should discuss any validation or cross-checking measures taken to ensure consistency between the two methods. The purpose for a 50/50 ratio between 3-day food records and 24-hour recalls might lead to inconsistencies in data quality and comparability.

Response 4: We agree with the reviewers that recall biases cannot be excluded as we indeed used also 24-hour dietary recalls to assess protein intake. In the study protocol, we aimed to collect dietary data from the 1-3 days prior to the uremic toxin determination. Although 3-day food records are considered the “gold standard” against other dietary assessment methods, it is widely accepted that 3-day food records are convinced labor intensive by patients and caregivers, face incomplete/selecting recording and requires literacy (summarized by McAlister et al. 2020). This is also the reason why we decided to alternate between the two techniques in a 50:50 ratio and to substitute the 3-day food records to 24 hours recall in case 3-day food records were not brought to the clinic. To minimize the impact of known shortcomings (i.e. interview bias, inaccurate estimation of portion size) by 24-hour recalls, the recalls were always performed according to a detailed protocol, i.e. by skilled and trained dieticians with the aid of standardized food models, a color photo atlas with choice between varying portion sizes and their corresponding weight of different food groups (Portiegroottes boek, Valetudo Consulting, third edition, March 2014), and a manual for the conversion of household measures to weight equivalents and standardized quantification of food items. The use of 24 hours recall has also several advantages that should also be taken into account: i.e. minimal participant burden, no literacy skills required, and cooking methods and cultural eating habits can be included. Of note, alternative strategies for assessment of dietary intake such as food frequency questionnaires, tend to overestimate daily estimated dietary intake in children were therefore not considered as a valuable alternative to 24 hours recall (summarized by McAlister et al. 2020). We agree with the reviewer that adding data on validity or reliability of our standardized 24 hours recalls versus the 3-day food records would strengthened the current study. This data is, however, not incorporated in our original study set-up. To address this shortcoming, we have made following modifications in the manuscript (limitation & methods section):

Page 7, line 203-208: Although we acknowledge that recall biases by the use of 24h recalls in our design cannot be excluded, the incorporation of the standardized 24h recall alternative have allowed us to balance the inherent disadvantages of 3-days food records in this observational study (i.e. high burden, incomplete recording, necessity of literacy skills).

Page 8, line 287:In order to minimize the impact of the known shortcomings of 24h recalls (i.e. interview bias, inaccurate estimation of portion size), the 24h recalls were performed according to a detailed protocol, i.e. by a selected number of skilled and trained dieticians with a standardized food models and a food photo album (Portiegroottes boek, Valetudo Consulting, 3rd edition, March 2014) were utilized, along with a manual for the conversion of household measures to weight equivalents[35, 36].”

Comments 5: Collecting dietary intake data every three months might not capture short-term dietary variations that could influence uremic toxin levels. More frequent dietary assessments could provide a complete understanding of the relationship between diet and uremic toxin variability.

Response 5: We thank the reviewer for raising this concern. However, we would like to highlight that all dietary assessments were collected 1-3 days prior to the sampling of protein-bound uremic toxins, to ensure the close and standardized relation between diet and plasma samples that we aimed for. Moreover, all patients were sampled during steady state, i.e. away for intercurrent viral/bacterial illnesses that could have affected appetite. Although we acknowledge that collecting dietary intake and uremic toxins more frequently is of interest to understand the relationship between diet and uremic toxin variability, we are dealing here with a pediatric population in which the number of additional blood samples (on top of routine care) and study visits significantly should be balanced with what is feasible for the patients. To explain this shortcoming more in detail, we have changed the method section accordingly:

Page 8, line 289: Structured 3-day diary templates were completed 3 days prior to the visit and reviewed by a trained dietician in face-to-face interviews. The 3-day food record was substituted by a 24h recall in case parents/patients failed to fill it out or forgot to bring it to the consult, so that dietary data could be coupled to plasma levels from the same day to ensure a standardized relation between diet and plasma samples.

5. Additional clarifications

Not applicable

Round 2

Reviewer 2 Report

Comments and Suggestions for Authors

The authors have made substantial changes in several part of the paper and addressed the reviewers’ comments. This manuscript may be accepted for publication.

Author Response

We thank the reviewer for the response.